# Clinical Utility of Three-Dimensional Speckle-Tracking Echocardiography in Heart Failure

**DOI:** 10.3390/jcm11216307

**Published:** 2022-10-26

**Authors:** Lang Gao, Yixia Lin, Mengmeng Ji, Wenqian Wu, He Li, Mingzhu Qian, Li Zhang, Mingxing Xie, Yuman Li

**Affiliations:** 1Department of Ultrasound Medicine, Union Hospital, Tongji Medical College, Huazhong University of Science and Technology, Wuhan 430022, China; 2Clinical Research Center for Medical Imaging in Hubei Province, Wuhan 430022, China; 3Hubei Province Key Laboratory of Molecular Imaging, Wuhan 430022, China; 4Shenzhen Huazhong University of Science and Technology Research Institute, Shenzhen 518057, China; 5Tongji Medical College and Wuhan National Laboratory for Optoelectronics, Huazhong University of Science and Technology, Wuhan 430022, China

**Keywords:** three-dimensional, speckle-tracking echocardiography, heart failure, clinical utility

## Abstract

Heart failure (HF) is an extremely major health problem with gradually increasing incidence in developed and developing countries. HF may lead to cardiac remodeling; thus, advanced imaging techniques are required to comprehensively evaluate cardiac mechanics. Recently, three-dimensional speckle-tracking echocardiography (3D-STE) has been developed as a novel imaging technology that is based on the three-dimensional speckle-tracking on the full volume three-dimensional datasets. Three-dimensional speckle-tracking echocardiography allows a more accurate evaluation of global and regional myocardial performance, assessment of cardiac mechanics, detection of subclinical cardiac dysfunction, and prediction of adverse clinical events in a variety of cardiovascular diseases. Therefore, this review summarizes the clinical usefulness of 3D-STE in patients with HF.

## 1. Introduction

Heart failure (HF) is defined as a clinical syndrome with typical symptoms (e.g., breathlessness and fatigue), accompanied by signs (e.g., increased jugular venous pressure and peripheral oedema) caused by structural and/or functional cardiac abnormality, resulting in a reduced cardiac output and/or increased intracardiac pressures at rest or during stress [1]. According to the latest statistics, HF involves approximately 40 million people worldwide [2], approximately 4.5 million patients in China [3], 1.3–23 million people in India, and 6.2 million adults in the United States [4,5]. HF patients have a long-standing subclinical period in which irreversible left ventricular (LV) remodeling and dysfunction occurs [1,6,7]. Early recognition of subclinical cardiac dysfunction may be helpful to slow the progression or prevent the development of HF [8]. Thus, early and accurate assessment of myocardial function in HF patients is essential for risk stratification, clinical decision-making and improvement of prognosis.

Currently, echocardiography remains the first-line imaging modality of assessment of cardiac structure and function, and it offers significant value in predicting HF events [9]. Although LV ejection fraction (LVEF) is the most commonly used echocardiographic parameter of cardiac function, it reflects only a relative volume change and is not a measure of contractility [10,11,12]. Recently, strain parameters using two-dimensional speckle-tracking echocardiography (2D-STE) have been shown to be more sensitive than LVEF for the assessment of early myocardial dysfunction [10,13,14,15]. Therefore, the HF guideline published by the ESC recommended myocardial strain as a sensitive index for the early detection of subclinical cardiac dysfunction [16]. However, the heart contracts in different directions and its motion is a three-dimensional (3D) phenomenon. Thus, the accurate assessment of cardiac function requires a 3D imaging modality. More recently, 3D speckle-tracking echocardiography (3D-STE) has been developed as a novel imaging technology that is based on the 3D speckle-tracking on the full volume 3D datasets, overcoming the intrinsic limitations of 2D-STE. Three-dimensional speckle-tracking echocardiography allows a more accurate evaluation of global and regional myocardial performance, assessment of cardiac mechanics, detection of subclinical cardiac dysfunction, and prediction of adverse clinical events in a variety of cardiovascular diseases. Consequently, the purpose of this review was to summarize the clinical usefulness of 3D-STE in patients with HF.

## 2. Evolution of Strain Imaging

Strain represents myocardial deformation that occurs during the cardiac cycle. Strain = (L_t_ − L_0_)/L_0_, where L_t_ is myocardial length at time t and L_0_ is original myocardial length at end-diastole. Strain is not affected by the movement of the entire heart [11]. Cardiac magnetic resonance (CMR) tagging was first used for early myocardial strain measurements in the late 1980s, whereas echocardiographic tissue Doppler imaging (TDI) measured one-dimensional strain in the 1990s. With the development of echocardiographic technology, 2D-STE was applied to cardiac function research in the early 2000s [13,17]. More recently, 3D-STE is a new technique developed on the basis of real time 3D echocardiography (RT-3DE) and speckle-tracking echocardiography (STE), overcoming the limitations of 2D-STE, and it can accurately and objectively evaluate the global and regional cardiac function.

## 3. Tissue Doppler Imaging

TDI is used to measure myocardial tissue velocities along the scan lines in one dimension and calculates strain parameters (e.g., regional strain, strain rate, and LV torsion) on this basis [18,19]. In all non-invasive imaging modalities, TDI allows a very rapid qualitative assessment of single myocardial segment and provides the unparalleled highest temporal resolution. However, the main limitations of this technique, including angle dependency, poor signal-to-noise ratio, and the effects of adjacent myocardial contraction and cardiac translational movements, limit the wide application of TDI in clinical practice [6,20,21].

## 4. Two-Dimensional Speckle-Tracking Echocardiography

Two-dimensional speckle-tracking echocardiography allows automatically tracking the motion trajectory of the myocardium frame by frame throughout the cardiac cycle by identifying the position and motion of speckles in the two-dimensional (2D) images, and parameters (e.g., myocardial velocity, strain, strain rate, rotation, and torsion) are obtained by post-processing [11,17,22]. Two-dimensional speckle-tracking echocardiography has advantages over traditional TDI. It allows a non-Doppler angle-independent analysis and is not influenced by adjacent myocardial segments and cardiac motion; therefore, feasibility, reproducibility, and accuracy of the 2D-STE data appear to be superior to traditional echocardiographic parameters [13,19,23,24,25]. A large number of studies have demonstrated that 2D-STE is a powerful technique for myocardial function quantification and can provide valuable diagnostic and prognostic information in clinical practice [26,27,28,29,30,31]. However, 2D-STE has several inherent limitations: it appears to work best at frame rates between 50 frames/s and 70 frames/s and may lead to under-sampling with tachycardia; whereas an increase in frame rate is detrimental to spatial resolution; inherent disadvantages of its 2D characteristics include the failure to track speckles that move out of the 2D spatial planes due to cardiac motion (through-plane phenomenon), and foreshortened views difficult to accurately assess myocardial deformation [11,20,26,27,32,33,34]. To overcome these shortcomings and comprehensively evaluate the myocardial mechanics from a 3D perspective, 3D-STE has been developed and introduced into the research and clinical arena.

## 5. Three-Dimensional Speckle-Tracking Echocardiography

LV myocardium consists of three distinct myocardial layers that contract simultaneously in different directions. Right ventricular (RV) myocardial fibers have a complex 3D arrangement (a superficial circumferential layer, a deep longitudinal layer, and a poorly developed mid-wall circumferential layer) with longitudinal orientation. Hence, the accurate assessment of cardiac function requires 3D analysis [17,35]. Three-dimensional speckle-tracking strain represents contraction and expansion of a pair of set points on the longitudinal, circumferential, and radial dimensions, as well as twists or untwists along the long axis. During cardiac systole, the longitudinal and circumferential myocardial shortening are expressed as negative strain and radial myocardial thickening as positive strain [18,36]. RV function has been assessed using RV-free wall strain only in many studies, as the septal strain contributes significantly to both LV and RV systolic function [35].

Three-dimensional speckle-tracking echocardiography is based on STE combined with RT-3DE, which is a speckle-tracking technique applied on RT-3DE. Three-dimensional speckle-tracking echocardiography tracks 3D motion trajectory of myocardial specific markers between successive volume frames within region of interest (ROI) frame by frame (3D template-matching algorithm) in full-volume 3D gray scale images, thereby derivating various deformation parameters of the myocardium throughout the entire cardiac cycle [17,35,36]. For the current 3D template-matching algorithm, electrocardiographic recording is vital because the acquisition of volumetric frames is timed using each RR interval of the electrocardiogram [8]. During the acquisition of 3D full volume, the ultrasound probe should correctly include all LV myocardium in the wide-angle pyramidal volume image at the apical position. The software then divides the acquired images into transverse sections and 2D apical, and automatically identifies and outlines the endocardial borders during the entire cardiac cycle so as to obtain parameters of myocardial deformation such as torsion, and global and segmental 3D strain [14,20,27] (Figure 1). Similarly, the software also automatically determines the RV endocardial borders, tracks the motion of the RV wall, and acquires the longitudinal strain of the RV-free wall and septum (Figure 2).

An accurate assessment of cardiac function requires 3D imaging methods, as myocardial movement is essentially a 3D phenomenon. Three-dimensional speckle-tracking echocardiography fuses advantages of 2D-STE and RT-3DE and has a better correlation with CMR, overcoming the limitation of 2D-STE that did not simultaneously measure all 3D displacement components of myocardial speckles in one cardiac cycle [28,37]. Three-dimensional speckle-tracking echocardiography enables omnidirectional tracking of all myocardial speckles in 3D space simultaneously, creating virtual 3D models of the cardiac chambers, detecting volume changes in the chambers during the cardiac cycle, allowing to avoid shortened images, and circumventing the errors caused by heart rate variability [8,22,38]. It also can comprehensively assess global and segmental myocardial function from the pyramidal dataset [28,29,39]. Notwithstanding, 3D-STE theoretically confers outstanding advantages over 2D-STE, the major weaknesses of this novel imaging modality include: low temporal and spatial resolution, tracking may be impaired if the thickness of ROI is too wide, and strain variability may be increased if the thickness of ROI is over-focused; strong dependence on the frame rates and image quality, which are vital determinants for remaining tracking quality, accurate edge detection, and strain evaluation; optimization of tracking requires operators to manually adjust contours; high vendor dependency need to account for data inconsistencies across vendors; offline analysis with extended examination time; and the effect of motion artifact. These drawbacks hamper the widespread use of 3D-STE in clinical practice [20,30,31,40,41,42,43]. Recently, the application of 3D-STE to assess cardiac function and prognosis in patients with cardiovascular diseases has become a hot topic in clinical research.

## 6. Three-Dimensional Speckle-Tracking Echocardiography in HF Patients

### 6.1. Left Ventricular Global Systolic Function

HF is defined based on the heart pump impairment (LV systolic dysfunction, LV diastolic dysfunction, RV systolic dysfunction, etc.) [44]. Three-dimensional speckle-tracking echocardiography can quantitatively evaluate LV systolic function by systolic strain parameters (e.g., longitudinal strain, circumferential strain, radial strain, and area strain) [17,27,32,45,46]. Sun et al. used 3D-STE to measure LV global strain (GS), global longitudinal strain (GLS), global circumferential strain (GCS), and global radial strain (GRS) in 65 HF patients and found that the above strain parameters were progressively decreased from HF with preserved ejection fraction (HFpEF) to HF with reduced ejection fraction (HFrEF) compared with normal controls. The incidence of impaired GLS, GCS, and GRS in HFpEF patients were 82%, 54%, and 36%, respectively. Thus, 3D-STE is of great value in assessing cardiac function in HF patients [47]. In a study by Li et al., GLS, global area strain (GAS), and GRS were impaired in patients with aortic stenosis and normal LVEF compared with healthy subjects (GLS −14.3% vs. −19.3%, *p* < 0.001). Pressure overload caused by aortic valve stenosis has a greater effect on GLS. Three-dimensional strains are useful to early detect cardiac dysfunction in patients with aortic stenosis and preserved LVEF [48].

Opposing rotations of the LV base and apex during systole results in LV torsion, a key parameter in cardiac function. Nagata et al. investigated which strain component was the robust predictor of major adverse cardiac events (MACEs) in 104 patients with asymptomatic severe aortic stenosis and LVEF >50% by 2D-STE and 3D-STE. In contrast to the patients without MACE, 3D GLS, 3D GRS, and 2D GLS were significantly lower in patients with MACE and had significant predictive power for MACE (3D GLS AUC = 0.78, cut-off value = −14.5%; 3D GRS AUC = 0.66, cut-off value = 39.0%; 2D GLS AUC = 0.62, cut-off value = −17.0%; all *p* < 0.01). Three-dimensional GLS was the most powerful independent predictor of MACE. Three-dimensional speckle-tracking echocardiography allows stratification of asymptomatic patients with severe aortic stenosis and preserved LVEF at high risk of MACE and supports early intervention and improves long-term outcomes [49]. Rady et al. sought to investigate LV torsion derived from 3D-STE as a marker of HF requiring hospitalization in patients with non-ischemic dilated cardiomyopathy (DCM) and revealed that the patients with a reduced LV torsion (<0.59 degrees/cm) had higher risk of HF hospitalization. Intriguingly, they observed a significantly reduction in LV torsion in patients with cardiac events than in those without cardiac events during follow-up. The evaluation of LV torsion may be of greater importance in the future first-line workup of patients with HF [50]. Luo et al. compared 44 patients with non-ST-elevation myocardial infarction (MI) with healthy controls using 3D-STE before and after percutaneous coronary intervention. Their findings showed that GLS, GCS, and LV systolic torsion were significantly decreased in HF patients with MI. GLS and LV systolic torsion significantly improved in patients after percutaneous coronary intervention. Interestingly, LVEF had a positive correlation with LV torsion, but negative correlations with LV GLS and LV GCS [29]. Three-dimensional speckle-tracking echocardiography provides a brand-new tool for evaluating LV torsion in HF patients, while GLS is the most robust deformation marker among various 3D-STE indices [10,12]. The superiority of GLS over other strain parameters include: superior resolution of images obtained in the axial plane; and in the non-hypertrophic heart, the amount of myocardial tissue in the apical long-axis view is more than in the short-axis view [24,29,40,51].

GLS reduction can be used as a prognostic marker of adverse cardiovascular outcomes of HF because it reflects the gradual development of LV dysfunction in HF patients [13,44]. In a retrospective investigation involving outcome-related risk markers in 270 patients with ST-elevation acute MI treated with reperfusion therapy, the primary endpoint events (all cardiac deaths, HF hospitalizations) were strongly correlated with 2D and 3D GLS than the other variables (2D GLS hazard ratio (HR) = 1.39, 95% CI 1.23–1.59, *p* < 0.0001; 3D GLS HR = 1.56, 95% CI 1.39–1.77, *p* < 0.0001). Moreover, models using 3D GLS were found to predict adverse outcomes stronger than that using 2D GLS. Three-dimensional GLS of >−11.1% was associated with primary endpoint events confirmed by Kaplan–Meier survival curves. The addition of 3D GLS to the base model (age + log BNP + infarct size) improved risk stratification in ST-elevation acute MI patients (net reclassification index = 0.5519%, *p* = 0.012). Three-dimensional GLS was a useful parameter to predict long-term prognosis in ST-elevation acute MI patients and had a significant incremental effect [52]. Therefore, reduced GLS was the independent predictor of HF outcomes. This conclusion was similarly corroborated by another original study recently published by the same group of authors [53]. Altman et al. assessed LV global systolic function in 147 patients using 2D-STE and 3D-STE. GLS, GRS, GCS, and area strain (AS) showed good accuracy in the detection of 2D LVEF < 55%, with AS indicating superiority over GRS and GCS but not GLS [54]. Current findings regarding assessment of LV global systolic function in HF patients using 3D-STE are presented in Table 1.

### 6.2. Left Ventricular Regional Systolic Function

Ischemic heart disease has a worse prognosis than patients with non-ischemic heart disease and is the most common cause of HF. Therefore, evaluation of regional myocardial function is crucial for better treatment of ischemic HF patients [28,55]. Echocardiography is considered to be the preferred method for cardiac function evaluation in acute MI [37,55,56]. The ability of 3D-STE to estimate actual 3D myocardial deformation may provide a good view of regional myocardial mechanics for sonographers [36,57,58]. The first study to evaluate 3D-STE technique for measuring regional wall motion indexes by Maffessanti et al. found that all 3D-STE indexes (segmental rotation, longitudinal and radial displacements, as well as longitudinal, circumferential, and radial strain) were reduced in the abnormal myocardial segments [59]. A study by Zhu et al. investigating 26 patients with acute MI using 3D-STE and CMR indicated that MI of different infarct sizes can be distinguished by 3D-STE, and LV GLS, GRS, and GCS correlated with the infarct area of myocardium measured by CMR (r = 0.81, −0.71, 0.86, respectively; all *p* < 0.01). Transmural infarct segments displayed markedly lower longitudinal, radial, and circumferential strains than normal segments [60]. Three-dimensional speckle-tracking echocardiography might provide a means to accurately determine the site of transmural infarct segments [60].

Due to the myocardial structural distribution characteristics of longitudinal endocardial fibers and circumferential mid-myocardial fibers, transmural MI may be represented by reductions in both longitudinal and circumferential strains [17,37]. AS, a combination of both longitudinal and circumferential strain, was shown to be an independent predictor of all-cause mortality or rehospitalization in HF and one of the sensitive parameters of early cardiac dysfunction, allowing accurate assessment of regional wall motion abnormalities [17,30,31]. Kleijn et al. demonstrated that AS was a promising index to quantitative evaluation of LV regional function and identified regional wall motion abnormalities in 114 consecutive patients with various heart diseases [61]. The specificity, sensitivity, and accuracy of AS in distinguishing abnormal from normal segments were more than 90% [62]. Li et al. applied 3D-STE to evaluate the LV remodeling of 61 patients with MI and indicated that the regional AS was significantly lower in the infarcted segments than those in the non-infarcted segments (−22.5 ± 10.9% vs. −40.5 ± 7.8%, *p* < 0.001), and the cut-off value of segmental AS (<−23%) well-differentiated regional MI had a specificity of 75.9% and a sensitivity of 75.1% (AUC = 0.82; 95% CI 0.77–0.86, *p* < 0.0001). They proved that 3D-STE can provide a new method for clinical assessment of the LV regional function in patients with MI [62]. However, several reports were contradictory when using 3D-STE to differentiate ischemic and non-ischemic etiologies in HF patients. Vachalcova et al. investigated 40 patients with HF and documented that LV apical rotation (4.9 ± 3.5° vs. 2.3 ± 2.4°, *p* = 0.0022) in HF patients with ischemic etiology significantly higher than in those non-ischemic etiology [28]. The results of another observation by Aly et al. suggested that ischemic and non-ischemic etiology of HF might be distinguished with the aid of 3D-STE. No significant differences were observed in global or regional strain parameters between the two groups, except for lower twist in the non-ischemic group (4.9 ± 3.3°vs 6.4 ± 3.2°, *p* = 0.03) [63]. In conclusion, for HF patients with ischemic etiology, 3D-STE has important clinical value in accurately and quantitatively assessing regional wall motion abnormalities, determining the extent of MI and transmural involvement. Table 2 summarizes current findings regarding assessment of LV regional systolic function in HF patients using 3D-STE.

### 6.3. Left Ventricular Diastolic Function

LV diastolic dysfunction is the results of cardiac pressure and volume overload, and further develops into HF. Therefore, LV diastolic function evaluation is considered a pivotal information for clinical diagnosis and prognosis of HF patients [44]. Evaluation of LV filling pressure is indispensable in the diagnosis and treatment of primary systolic and diastolic HF. Tatsumi et al. investigated 125 patients with LVEF of 40 ± 17% who underwent 3D-STE and right heart catheterization, and confirmed that 3D-STE can be used to assess the LV diastolic function. The ratio of peak early diastolic transmitral velocity and early diastolic area change rate (E/E-ACR, AUC = 0.82, *p* < 0.001; cut-off value ≥ 94 cm) was a powerful predictor of elevated LV filling pressure [64]. A study published recently evaluated LV diastolic function by applying 3D-STE in 112 patients with coronary heart disease. Significantly decreased GLS, GCS, GRS, GAS, and LVEF in patients with diastolic dysfunction were observed in comparison with the control group [65]. Three-dimensional speckle-tracking echocardiography can provide imaging evidences for the evaluation of LV diastolic function.

Patients with HF usually present with exercise intolerance and the gold standard for assessing clinical cardiopulmonary function is cardiopulmonary exercise testing (CPET) [66]. Combined with 3D-STE and CPET data from 156 subjects with different levels of cardiac function, Li et al. found that the amount of 25% duration untwisting (25% untwist) was positively correlated with peak oxygen uptake (r = 0.41, *p* < 0.001) and negatively correlated with the CO_2_ equivalent slope (r = −0.49, *p* < 0.001). A total of 64% of the variation in peak oxygen uptake and 17.6% of the variation CO_2_ equivalent slope were explained by 25% untwist. A total of 25% untwist value was a good indicator of LV early diastolic dysfunction, which is an independent predictor for examining aerobic exercise capacity. Cardiac function of HF patients with dyskinesia can be evaluated as an aid by revealing the relationship between cardiac diastolic function at rest and CPET [66].

### 6.4. Cardiac Resynchronization Therapy

Cardiac resynchronization therapy (CRT), an important treatment for patients with advanced HF as it improves the pattern of myocardial contraction by placing strategically positioned biventricular pacemaker [67]. Although LV mechanical dyssynchrony is associated with improved cardiac function after CRT, responses to post-CRT therapy are hard to predict and can be absent in as many as one-third of patients. The selection of optimal candidates for CRT can be assisted by echocardiographic assessment of LV mechanical dyssynchrony [17,68,69]. M-mode, pulsed Doppler and TDI methods have yielded mixed results in predicting CRT response, while STE has emerged as a new method for predicting treatment outcomes in CRT patients through the measurement of myocardial deformation [32,69]. Three-dimensional speckle-tracking echocardiography can show the specific status of each myocardial segments in HF patients during the same cardiac cycle, which helps to better analyze ventricular mechanical dyssynchrony.

In a study investigating the evaluation of LV systolic mechanical synchrony in 52 dilated cardiomyopathy (DCM) patients and 55 healthy controls, the standard deviation of time to negative peak strain (3DS-SD) and time to regional minimum volume of 16 segments (SDI) derived from 3D-STE were used to explore the LV dyssynchrony [70]. Kang et al. demonstrated that LV 3D strain, 3D Longitudinal strain, 3D radial strain, and 3D circumferential strain had significant correlations with LVEF (r^2^ = −0.94, −0.91, −0.89, 0.93, respectively; all *p* < 0.01). After CRT, 3DS-SDs (12.99 ± 0.039 to 11.53 ± 0.045, *p* < 0.05) and SDIs (10.85 ± 0.037 to 9.50 ± 0.046, *p* < 0.05) improved significantly. The assessment of LV dyssynchrony using 3D-STE might be critical for selecting HF patients who can benefit from CRT [70]. Szulik et al. revealed that 3D GLS (AUC = 0.756, *p* < 0.05, cut-off value > −9.52%, 78% sensitivity, 80% specificity), RS (AUC = 0.739, *p* = 0.086; cut-off value = 20%, 78% sensitivity, 80% specificity), and AS (AUC = 0.733, *p* < 0.05; cut-off value > −13.5%, 67% sensitivity, 80% specificity) can be used to select resynchronization candidates in the HF group according. Three-dimensional speckle-tracking echocardiography is applicable in the evaluation of both patients with HFpEF and HFrEF and helps to assess the CRT candidates and response [71]. Zhu et al. assessed LV myocardial synchrony through the slope and the coefficient of determination (R^2^ -S/D coupling) to explore the value of strain-volume loops using 3D-STE analysis in predicting CRT response in HF patients. They found that Midseptal R^2^-S/D coupling at baseline was an independent predictor of CRT response, with a recommended cut-off value of 0.55, a sensitivity of 89%, and a specificity of 77%, and deduced that strain-volume loops provided important information for predicting CRT response. Thus, 3D-STE is helpful for improving the selection of CRT patients and may provide significant information to predict response to CRT [72].

### 6.5. Left Atrial Function

The main function of left atrium is to regulate LV filling [73]. The thin walls and unique anatomy of the left atrium determine that it is extremely sensitive to injury [74]. Therefore, it is not surprising that the evident changes in left atrial (LA) function occur at the earliest stages of LV diastolic dysfunction [75,76]. The assessment of LA function may provide great values in the diagnosis, risk stratification, and efficacy monitoring of LV diastolic function in patients with HFpEF [77,78,79,80]. Three-dimensional speckle-tracking echocardiography can analyze the LA deformation of all segments to accurately assess LA function without being affected by heart rhythm [81,82]. Indices of LA volume have been reported as markers of poor prognosis in HF patients. Tsujiuchi et al. followed 514 patients with cardiovascular diseases for 720 ± 383 days and obtained the LA maximal, minimal volume indices and LA emptying fraction (LAEmpF) using 2D echocardiography and 3D-STE. The results showed that the cut-off values measured by 3D-STE strongly predicted major cardiovascular events by Kaplan–Meier survival analysis (*p* < 0.001) [83]. Importantly, 3D-STE-derived LAEmpF (AUC = 0.82, *p* < 0.0001; cut-off value < 0.420) was an independent predictor of hospitalization in HF patients, providing higher prognostic power in future MACEs than all models using 2D parameters. Thus, measurement of LA parameters calculated by 3D-STE is clinically useful and feasible [83]. The same group of authors followed 264 patients to record major cardiovascular events, and reported that 2D LV longitudinal strain (HR = 5.37, *p* < 0.001), 3D LA longitudinal strain (HR = 5.57, *p* < 0.001), and 3D LAEmpF (HR = 6.59, *p* < 0.001) showed higher hazard ratios than other indices. Three-dimensional LAEmpF + base model (AUC = 0.837, *p* = 0.02; cut-off value < 0.330) displayed incremental prognostic values in major cardiovascular events [84]. Three-dimensional speckle-tracking echocardiography combined with RT-3DE can effectively assess the LA function by quantifying LA strain. During the evaluation of left atrial function changes in patients with HFpEF by combining 3D-STE with RT-3DE, Liu et al. found that LA reservoir, conduit, and pump function in 43 HFpEF patients were significantly reduced, and strains of LA middle level (systolic longitudinal peak strain and pre-contraction longitudinal peak strain) were powerful parameters for assessing LA function with satisfactory reproducibility [85].

LV diastolic dysfunction is the underlying mechanism of both atrial fibrillation (AF) and HFpEF, and HF is often accompanied by AF comorbidities [82,86,87,88]. Mochizuki et al. followed 42 patients with paroxysmal AF before catheter ablation for 441 ± 221 days and determined systolic LA global peak longitudinal (GLSs), circumferential (GCSs), and area strain (GASs) during systole and before atrial contraction by 3D-STE and 2D-STE, respectively [89]. Standard deviation of LA regional strain times to peak was considered as indicators of LA asynchrony. The authors found that systolic 3D GCSs (HR = 0.91, *p* = 0.03), 3D GASs (HR = 0.95, *p* = 0.01), and age (HR = 1.08, *p* = 0.04) may predict the recurrence of AF. Moreover, 3D LA strain was a better predictor of AF recurrence after catheter ablation than 2D or other predictors. Three-dimensional global area strains (AUC = 0.71, *p* = 0.029; cut-off value > 28.9%) may help to predict the risk of AF recurrence and select patients with improved AF after catheter ablation [89]. A study by Esposito et al. explored LA function in 48 patients with hypertensive and paroxysmal AF, and showed that impairment of LA strain and strain rates were observed in both hypertensive and paroxysmal AF groups compared with control subjects. Three-dimensional strain provided more significant LA functional information and LA strain was an independent predictor of poor outcomes in patients with paroxysmal AF [90]. Table 3 illustrates current findings regarding assessment of LA function in HF patients using 3D-STE.

### 6.6. Right Ventricular Function

RV function assessment is particularly vital in clinical practice, considering the fact that RV failure is the leading cause of death in patients with various cardiovascular diseases, including pulmonary arterial hypertension, RV MI, congenital heart disease, and HF [91,92]. The development of RV failure is due to pulmonary hypertension leading to increased afterload and tricuspid regurgitation with volume overload [93]. However, the right ventricle is difficult to accurately analyze through conventional echocardiography owing to its position in the chest, complex 3D anatomy, and non-geometric shape [22,35]. Three-dimensional speckle-tracking echocardiography developed in recent years is not affected by geometric assumptions and provides a new method for objective quantification of RV volumes, global and regional function, and mechanics [20,65,91,92,93].

In a retrospective cross-sectional study with 106 patients with underlying heart diseases who underwent both 3D echocardiography and CMR imaging within one month, researchers found that inlet area strain and outflow circumferential strain from 3D-STE were independently correlated with RV ejection fraction (RVEF) assessed by CMR, as a gold-standard imaging technique for quantifying cardiac function. Three-dimensional speckle-tracking echocardiography might be a promising method for assessing RV function [94]. Two-dimensional RV-free wall longitudinal strain (RVFWLS) has been considered as an independent predictor of adverse outcomes in HFpEF patients. Meng et al. investigated the predictive value of 3D-STE and 2D-STE in 93 HFpEF patients during 17-month median follow-up [95]. Their findings showed that 3D-STE parameters (3D RVFWLS, 3D RVEF) and the 2D-STE parameter (2D RVFWLS) were independent predictors of poor prognosis (HR = 5.73,3.47,3.17; 95% CI 2.77–11.85,1.47–8.21,1.54–6.53; respectively, all *p* < 0.01). Furthermore, 3D RVFWLS and 3D RVEF had comparable predictive ability to 2D RVFWLS for adverse cardiac events in multivariate Cox analysis models (akaike information criterion (AIC) = 246,247,248; C-index = 0.75,0.76,0.74; respectively). Hence, 3D-STE displays the potential to identify patients with HFpEF at high risk of poor clinical outcomes [95]. In addition, the main causes of death in maintenance hemodialysis patients are cardiovascular disease, and RV dysfunction is associated with increased cardiovascular morbidity and mortality [96]. Sun et al. investigated the RV function in 80 patients with maintenance hemodialysis and 30 healthy subjects using 3D-STE. The RV parameters of 3D-STE (end-systolic and end-diastolic volumes, RVEF, and longitudinal strains) and traditional echocardiographic indices (RV end-systolic and end-diastolic areas, tricuspid annular systolic displacement, fractional area change, tricuspid annular peak systolic velocity, and index of myocardial performance) were compared between the two groups. In contrast to the healthy controls, RV longitudinal strain was significantly impaired in patients (all *p* < 0.001) [96]. Vitarelli et al. compared 73 patients with chronic pulmonary hypertension with 30 healthy subjects and showed that the impairments of 3D RVEF and RVFWLS were noted in patients with chronic pulmonary hypertension compared with controls [97]. Both 3D RVEF and RVFWLS had large AUC for identifying RV failure hemodynamics, and they (3D RVEF HR = 4.6, 95% CI 2.79–8.38, *p* = 0.004; RVFWLS HR = 5.3, 95% CI 2.85–9.89, *p* = 0.002; respectively) were independently associated with higher mortality in these patients [98]. Collectively, 3D-STE can quantitatively assess global and regional RV function and provide valuable diagnostic and prognostic information for HF patients.

### 6.7. Myocardial Fibrosis

End-stage Myocardial compensation is mostly characterized by collagen and fibrosis formation [98,99]. Patients with advanced HF have severely impaired cardiac function because of significant fibrosis [100]. Due to the irreversible nature of fibrosis, the treatment of advanced HF patients is very challenging, contributing to serious consequences including cardiac death [98,101]. Thus, early detection of myocardial deformation abnormalities is very important to assess myocardial fibrosis (MF).

MF was previously recognized by endomyocardial biopsy as an invasive method [75]. Recently, with the advent of the novel STE technique, it is possible to obtain information on the presence of MF by analyzing deformation parameters, forming a non-invasive assessment [101,102]. Wang et al. used 2D-STE and 3D-STE to predict LV MF in 75 DCM patients with advanced HF, and reported that 2D GLS, 3D GLS, GRS and tangential strain (3DTS) were lower in patients with severe MF than those with mild and moderate MF [103]. MF was closely related to 3D GLS (r = 0.72, *p* < 0.001), which was found to be more accurate in predicting MF than 2D GLS and 3DTS in multivariate linear regression models (R^2^ = 0.76,0.66,0.65; AIC = 331,352,354; respectively). Furthermore, the AUC of 3D GLS in identifying severe MF was significantly greater than those of other strain parameters (0.86 vs. 0.59–0.70; all *p* < 0.05; cut-off value = −9.7%). Three-dimensional GLS might be an excellent surrogate marker for evaluating LV MF in patients with DCM and advanced HF, and will be useful for risk stratification, determining a time for early clinical intervention and assessing efficacy [103]. Another original research published recently by the same group of authors, which enrolled 105 end-stage HF patients with heart transplantation, investigated the value of RV strain parameters derived from 2D-STE and 3D-STE for predicting RV MF. The results demonstrated that patients with severe RV MF had lower 3D RVFWLS than patients with mild and moderate RV MF [104]. Three-dimensional RVFWLS was highly correlated with RV MF (r = −0.72, *p* < 0.001) and correlated strongest with the MF degree (r = −0.72 vs. −0.21 to −0.53; all *p* < 0.05) compared with 2D-STE and other conventional RV function parameters. In addition, among all RV parameters involved in this study, 3D RVFWLS was the most accurate parameter (AUC = 0.90 vs. 0.24–0.80; *p* < 0.05; cut-off value = −9.46%) for detecting severe MF. Consequently, 3D RVFWLS may be the most reliable echocardiographic parameter to predict the degree of RV MF in end-stage HF patients, especially those who need serial follow-up or are unable to undergo CMR examinations [104].

## 7. Summary

HF, as the terminal stage in the progression of various cardiovascular diseases, has become a significant problem threatening public health worldwide. Therefore, accurate assessment of cardiac function has vital clinical value in definitive diagnosis, clinical decision-making, risk stratification, and serial follow-up of HF patients. Three-dimensional speckle-tracking echocardiography serves as a novel imaging technology and has been increasingly applied in the research and clinical areas due to its advantages of excellent accuracy, high efficiency, low cost, and 3D analysis. With the continuous improvements in STE technology, increasing resolution, and image quality, 3D-STE can be considered to be a promising tool in evaluating cardiac function and providing accurate diagnostic and prognostic information in clinical practice.

## Figures and Tables

**Figure 1 jcm-11-06307-f001:**
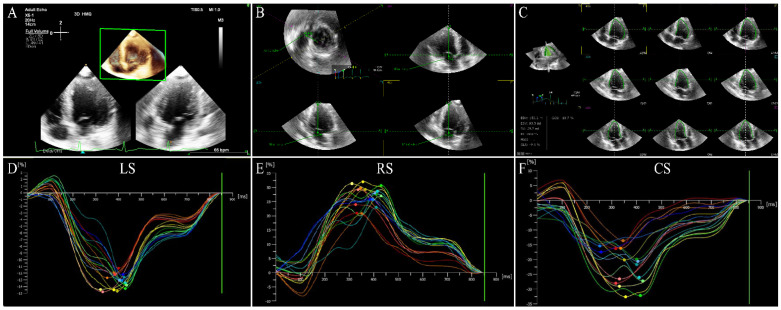
LV three−dimensional strain images in a HFrEF patient. (**A**) Three-dimensional image of LV−focused apical 4−chamber view. (**B**) Reference points setting. (**C**) LV endocardial border identification and tracking. (**D**−**F**) Longitudinal, radial, and circumferential strain of left ventricle. LV: left ventricular; LS: longitudinal strain; RS: radial strain; CS: circumferential strain; HFrEF: heart failure with reduced ejection fraction.

**Figure 2 jcm-11-06307-f002:**
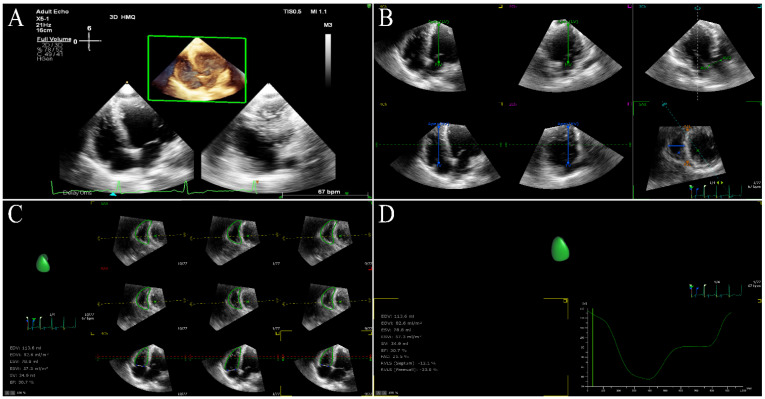
RV three−dimensional strain images in a HFrEF patient. (**A**) Three−dimensional image of RV−focused apical 4−chamber view. (**B**) Reference points setting. (**C**) RV endocardial border identification and tracking. (**D**) Longitudinal strain of RV-free wall and septum. RV: right ventricular.

**Table 1 jcm-11-06307-t001:** Assessment of LV global systolic function in HF patients using 3D-STE.

References	Sample Size	Age (Years)	Men, *n* (%)	LVEF (%)	Strain Parameters	Main Findings
Sun et al. [47]	65	62 ± 8 ^a^ 61 ± 9 ^b^	Not reported	59.5 ± 4.2 ^a^ 36.2 ± 5.3 ^b^	GS, GLS, GCS, GRS	GS, GLS, GCS, and GRS were progressively decreased from HFpEF to HFrEF compared with normal controls.
Li et al. [48]	34	55.6 ± 16.7	*19* (55.9)	59.1 ± 7.5	GLS, GAS, GRS	GLS, GAS, and GRS were impaired in the aortic valve stenosis group compared with healthy controls.
Nagata et al. [49]	104	78 ± 10	*43* (41.3)	60 ± 5	GLS, GRS	3D GLS, 3D GRS, and 2D GLS had significant predictive power for MACE. Three-dimensional GLS was the most powerful independent predictor of MACE.
Rady et al. [50]	91	53 ± 13	*74* (81.3)	33 ± 10	GLS, GCS, LV torsion	The patients with a reduced LV torsion (<0.59 degrees/cm) had higher risk of HF hospitalization.
Luo et al. [29]	44	62.3 ± 8.4 ^a^ 64.9 ± 4.8 ^b^ 61.1 ± 9.3 ^c^	*11* (78.6) ^a^ *7* (63.6) ^b^ *12* (63.2) ^c^	56.42 ± 4.61 ^a^ 43.89 ± 6.19 ^b^ 59.34 ± 2.20 ^c^	GLS, GCS, LV torsion	GLS, GCS, and LV systolic torsion were significantly decreased in HF patients with MI.
Iwahashi et al. [52]	270	65	*222* (82.2)	52	GLS, GCS, GRS, global principal strain	The model using 3D GLS was found to predict adverse outcomes stronger than those using 2D GLS. Three-dimensional GLS of >−11.1 was associated with primary endpoint events.
Iwahashi et al. [53]	248	64	*206* (83.1)	56 ± 12.1	GLS	The reduced 3D GLS at 1 year was an independent predictor for the primary endpoint events.
Altman et al. [54]	147	54 ± 15	Not reported	21–72	GLS, GRS, GCS, AS	GLS, GRS, GCS, and AS showed good accuracy in the detection of 2D LVEF < 55%, with AS indicating superiority over GRS and GCS but not GLS.

^a^ HFpEF group; ^b^ HFrEF group; ^c^ HF with normal ejection fraction group. LV: left ventricular; HF: heart failure; 3D-STE: three-dimensional speckle-tracking echocardiography; LVEF: left ventricular ejection fraction; GS: global strain; GLS: global longitudinal strain; GCS: global circumferential strain; GRS: global radial strain; HFpEF: heart failure with preserved ejection fraction; HFrEF: heart failure with reduced ejection fraction; GAS: global area strain; MI: myocardial infarction; 3D: three-dimensional; 2D: two-dimensional; AS: area strain; MACE: major adverse cardiac event.

**Table 2 jcm-11-06307-t002:** Assessment of LV regional systolic function in HF patients using 3D-STE.

References	Sample Size	Age (Years)	Men, *n* (%)	LVEF (%)	Strain Parameters	Main Findings
Maffessanti et al. [59]	32	59 ± 17	*20* (62.5)	Not reported	Longitudinal strain, circumferential strain, radial strain, rotation	All 3D-STE indexes were reduced in the abnormal myocardial segments.
Zhu et al. [60]	26	56.3 ± 11.1	*15* (57.7)	51.3 ± 5.73 ^a^ 44.69 ± 6.73 ^b^ 41.22 ± 9.29 ^c^	GLS, GCS, GRS, longitudinal strain, circumferential strain, radial strain	LV GLS, GRS, and GCS correlated with the infarct area of myocardium measured by CMR. Transmural infarct segments displayed markedly lower longitudinal, radial, and circumferential strains than normal segments.
Kleijn et al. [61]	114	59 ± 16	*67* (58.8)	51 ± 13	LV AS	AS was a promising index to quantitative evaluation of LV regional function and identify regional wall motion abnormalities.
Li et al. [62]	61	54.2 ± 9.2 ^d^ 53.2 ± 8.7 ^e^	*15* (60) ^d^ *21* (58.3) ^e^	44.7 ± 4.3	LV GAS, LV regional peak AS	Regional AS were significantly lower in the infarcted segments than those in the non-infrared segments.
Vachalcova et al. [28]	40	63 ± 9 ^f^ 64 ± 11^g^	*19* (95) ^f^ *15* (75) ^g^	29.0 ± 11.3 ^f^ 27.3 ± 7.5 ^g^	GLS, twist, LV apical rotation	LV apical rotation in HF patients with ischemic etiology is significantly higher than in those non-ischemic etiology.
Aly et al. [63]	120	63 ± 12 ^f^ 59 ± 14 ^g^	*65* (79) ^f^ *27* (61) ^g^	40 ± 8 ^f^ 37 ± 9 ^g^	Longitudinal strain, circumferential strain, radial strain, 3D strain, AS, LV twist	No significant differences were observed in global or regional strain parameters between the two groups, except for lower twist in non-ischemic group.

^a^ Small-size MI group; ^b^ moderate-size MI group; ^c^ large-size MI group; ^d^ remodeled group; ^e^ non-remodeled group; ^f^ ischemic etiology group; ^g^ Non-ischemic etiology group. CMR: cardiac magnetic resonance; other abbreviations as in Table 1.

**Table 3 jcm-11-06307-t003:** Assessment of LA function in HF patients using 3D-STE.

References	Sample Size	Age (Years)	Men, *n* (%)	LVEF (%)	Strain Parameters	Main Findings
Tsujiuchi et al. [83]	514	66 ± 15	*320* (62)	55 ± 16	Maximal LA volume index, minimal LA volume index, LAEmpF	3D-STE-derived LAEmpF was an independent predictor of hospitalization in HF patients, providing higher prognostic power than that measured by 2D echocardiography.
Tsujiuchi et al. [84]	264	65 ± 16	*159* (60)	55 ± 16	Maximal LA volume index, minimal LA volume index, LAEmpF, LA longitudinal strain, LA circumferential strain	2D LV longitudinal strain, 3D LA longitudinal strain, and 3D LAEmpF showed higher hazard ratio than other parameters. Three-dimensional LAEmpF displayed incremental prognostic values in major cardiovascular events.
Liu et al. [85]	43	55.7 ± 6.9 ^a^ 54.8 ± 7.4 ^b^	*11* (57.9) ^a^ *12* (50.0) ^b^	58 ± 10 ^a^ 60 ± 6 ^b^	Maximal LA volume index, minimal LA volume index, longitudinal strain, LA stiffness index	LA reservoir, conduit, pump function in HFpEF patients were significantly reduced, and strains of LA middle level were powerful parameters for assessing LA function.
Mochizuki et al. [89]	42	58 ± 10	*29* (69)	66 ± 7	LA GLS, LA GCS, LA GAS	Systolic 3D GCS and age may predict the recurrence of AF, and 3D LA strain was a better predictor of AF recurrence after catheter ablation than other predictors.
Esposito et al. [90]	48	59.78 ± 13.8 ^c^ 56.39 ± 7.58 ^d^ 57.5 ± 9.62 ^e^ 71.64 ± 3.96 ^f^	*8* (89) ^c^ *10* (56) ^d^ *9* (90) ^e^ *6* (55) ^f^	60.11 ± 4.46 ^c^ 60.39 ± 5.08 ^d^ 55.7 ± 3.86 ^e^ 60 ± 5.1 ^f^	LA GLS, LA GCS, LA GAS, strain rate	Impairment of LA strain and strain rates were observed in both hypertensive and paroxysmal AF groups compared with control subjects, and LA strain was an independent predictor of poor outcomes in patients with paroxysmal AF.

^a^ Maximal LA volume index < 34 mL/m^2^; ^b^ maximal LA volume index ≥ 34 mL/m^2^. ^c^ Patients with arterial hypertension (HT) and paroxysmal AF, without LV hypertrophy (LVH); ^d^ patients with HT without LVH; ^e^ patients with HT and LVH; ^f^ patients with paroxysmal AF, HT, and LVH. LA: left atrial; LAEmpF: left atrial emptying fraction; AF: atrial fibrillation; other abbreviations as in Table 1 and Table 2.

## Data Availability

Not applicable.

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
