# Peer review of "Clinical Utility of Three-Dimensional Speckle-Tracking Echocardiography in Heart Failure"

_jcm, 2022, doi:10.3390/jcm11216307_

Round 1

Reviewer 1 Report

After reading this review article regarding clinical utility of 3D speckle tracking echocardiography analysis in heart failure, I did not find any good take home message. 3D speckle tracking analysis is based on the 3D speckle tracking on the full volume 3D datasets, not 2D speckle tracking on 2D image extracted from 3D datasets. The results were mainly superficial, and figures and tables did not provide any meaningful information. Finally, in the section of myocardial fibrosis, the authors compared diagnostic value of both 2D RVFWLS and 3D RVFWLS for RV myocardial fibrosis. Unfortunately, TomTec 4D-RV Function does not provide 3D RV strains. It merely provides 2D RVFWLS on 2D image extracted from 3DE datasets.

Author Response

Response to Reviewer 1 Comments

Point 1: After reading this review article regarding clinical utility of 3D speckle tracking echocardiography analysis in heart failure, I did not find any good take home message. 3D speckle tracking analysis is based on the 3D speckle tracking on the full volume 3D datasets, not 2D speckle tracking on 2D image extracted from 3D datasets. The results were mainly superficial, and figures and tables did not provide any meaningful information. Finally, in the section of myocardial fibrosis, the authors compared diagnostic value of both 2D RVFWLS and 3D RVFWLS for RV myocardial fibrosis. Unfortunately, TomTec 4D-RV Function does not provide 3D RV strains. It merely provides 2D RVFWLS on 2D image extracted from 3DE datasets.

Response 1:We are grateful for your comment. We totally agree with your comment on the principles of 3D speckle tracking analysis. And we have revised it in the revised manuscript. (Lines19-20, 51-52, 108)

  We also agree with your viewpoint that TomTec 4D-RV Function merely provides 2D RVFWLS on 2D image extracted from 3DE datasets. However, reconstructed RV-focused four-chamber views extracted from 3DE datasets are not affected by foreshortening which is sometimes challenging using just 2D echocardiography. Therefore, we think that RVFWLS derived from TomTec 4D-RV Function may be better than that measured on 2D images.

Once again, we thank this reviewer for the extremely insightful and constructive comments and critiques.

Reviewer 2 Report

Dear Author,

A well done review paper covering all aspects of the new 3D speckle tracking echocardiography. I am particularly interested in the advances in early detection of HFpEF problems, as this is one of the unresolved problems of patients with aortic valve stenosis who need surgery but often wait too long. Perhaps you can elaborate on this a bit more. Otherwise, well done.

Author Response

Response to Reviewer 2 Comments

Point 1: A well done review paper covering all aspects of the new 3D speckle tracking echocardiography. I am particularly interested in the advances in early detection of HFpEF problems, as this is one of the unresolved problems of patients with aortic valve stenosis who need surgery but often wait too long. Perhaps you can elaborate on this a bit more. Otherwise, well done.

Response 1Thank you very much for your nice comments. We have added the additional clinical usefulness of 3D-STE in detecting early cardiac dysfunction in patients with aortic valve stenosis and preserved LVEF. (Lines165-169, 171-180)

Once again, we appreciate the expertise and insight of the reviewer and the very helpful comments and questions. 

Reviewer 3 Report

Gao et al wrote a review article on 3D speckle-tracking echocardiography (3D-STE).

Can authors comment more how the machine calculates strain for 3D speckle?

In line 142 – HF is not always systolic dysfunction. This should be revised.

The article is somewhat difficult to follow. Strain is known to predict cardiovascular outcomes and the authors showed that. I recommend to include clinical applications of how and when to use it.

Also include examples of 3D strain images for the readers.

The part on 3D strain for CRT doesn't give clinical guidance and the data behind it is not very strong.

Author Response

Response to Reviewer 3 Comments

Point 1: Gao et al wrote a review article on 3D speckle-tracking echocardiography (3D-STE). Can authors comment more how the machine calculates strain for 3D speckle?

Response 1We thank the reviewer for the valuable comment. We have added the calculation of 3D speckle-tracking strain in the revised manuscript. (Lines 58-60, Lines 97-107)

Point 2: In line 142 – HF is not always systolic dysfunction. This should be revised.

Response 2Thank you very much for your careful review. We have revised this issue in the revised manuscript. (Lines 155-156)

Point 3: The article is somewhat difficult to follow. Strain is known to predict cardiovascular outcomes and the authors showed that. I recommend to include clinical applications of how and when to use it.

Response 3We thank the reviewer for these valuable comments. We have added relevant information (diagnosis, cut-off value, risk stratification, monitoring the response to the treatment, etc.) in the revised manuscript. (Lines 171-180, 206-210, 247-249, 273-274, 338-342, 344-346, 363-365, 394-398, 436-438, 447, 449-450)

Point 4: Also include examples of 3D strain images for the readers.

Response 4Thank you for your comment. We have included the 3D strain echocardiographic images of patients with HF in the manuscript (Lines 123-131, Figure 1 and Figure 2). We 're sorry that we didn' t indicate it clearly in the figure legends. And we have revised it in the revised manuscript. (Lines 124, 129)

Point 5: The part on 3D strain for CRT doesn't give clinical guidance and the data behind it is not very strong.

Response 5Thank you very much for your valuable review. We have rewritten the section of 3D strain for CRT and further analyzed its clinical implications. (Lines 303-316)

Once again, we appreciate the insightful and constructive comments and questions of the reviewers.

Round 2

Reviewer 3 Report

Manuscript improved significantly.